# Divert More Attention to Vision-Language Tracking

**Mingzhe Guo**[1*]   **Zhipeng Zhang**[2*]   **Heng Fan**[3]   **Liping Jing**[1†]

[1]Beijing Key Lab of Traffic Data Analysis and Mining, Beijing Jiaotong University
[2]DiDi Chuxing, Beijing, China
[3]Department of Computer Science and Engineering, University of North Texas
{mingzheguo, lpjing}@bjtu.edu.cn, zhipeng.zhang.cv@outlook.com, heng.fan@unt.edu

## Abstract

Relying on Transformer for complex visual feature learning, object tracking has witnessed the new standard for state-of-the-arts (SOTAs). However, this advancement accompanies by larger training data and longer training period, making tracking increasingly expensive. In this paper, we demonstrate that the Transformer-reliance is not necessary and the pure ConvNets are still competitive and even better yet more economical and friendly in achieving SOTA tracking. Our solution is to unleash the power of multimodal vision-language (VL) tracking, simply using ConvNets. The essence lies in learning novel unified-adaptive VL representations with our modality mixer (ModaMixer) and asymmetrical ConvNet search. We show that our unified-adaptive VL representation, learned purely with the ConvNets, is a simple yet strong alternative to Transformer visual features, by unbelievably improving a CNN-based Siamese tracker by 14.5% in SUC on challenging LaSOT (**50.7%**→**65.2%**), even outperforming several Transformer-based SOTA trackers. Besides empirical results, we theoretically analyze our approach to evidence its effectiveness. By revealing the potential of VL representation, we expect the community to divert more attention to VL tracking and hope to open more possibilities for future tracking beyond Transformer. Code and models are released at https://github.com/JudasDie/SOTS.

## 1 Introduction

Transformer tracking recently receives a surge of research interests and becomes almost a necessity to achieve state-of-the-art (SOTA) performance [8, 52, 10]. The success of Transformer trackers mainly attributes to *attention* that enables complex feature interactions. But, is this complex attention the only way realizing SOTA tracking? Or in other words, *is Transformer the only path to SOTA?*

We answer **no**, and display a *Transformer-free* path using **pure** convolutional neural network (CNN). Different than complex interactions in visual feature by attention requiring more training data and longer training time, our alternative is to explore simple interactions of multimodal, *i.e.*, vision and language, through CNN. In fact, language, an equally important cue as vision, has been largely explored in vision-related tasks, and is not new to tracking.

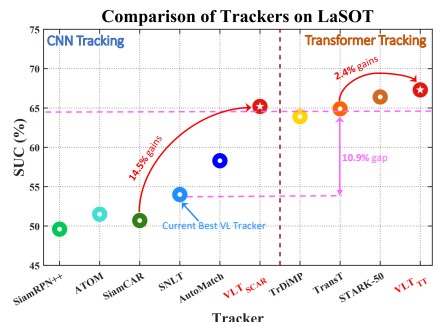

Figure 1: Comparison between CNN-based and Transformer-based trackers on LaSOT [17].

---

[*]Equal Contribution. [†] Corresponding author.
This work is co-supervised by Prof. Liping Jing and Dr. Zhipeng Zhang.

36th Conference on Neural Information Processing Systems (NeurIPS 2022).

Prior works [20, 34, 19] have exploited vision-language (VL) multimodal learning for improving tracking. However, the performance falls far behind current SOTAs. For instance on LaSOT [17], the gap between current best VL tracker [20] and recent Transformer tracker [8] is absolute 10.9% in SUC (see Fig. 1). *So, what is the bottleneck of VL tracking in achieving SOTA?*

**The devil is in VL representation.** Feature representation has been shown to be crucial in improving tracking [51, 61, 32, 24, 10]. Given two modalities of vision and language, the VL feature is desired to be *unified* and *adaptive* [40, 58]. The former property requires deep interaction of vision and language, while the latter needs VL feature to accommodate different scenarios of visual and linguistic information. However, in existing VL trackers, vision and language are treated *independently* and processed *distantly* until the final result fusion. Although this fusion may easily get two modalities connected, it does not accords with human learning procedure that integrates multisensory by various neurons before causal inference [3], resulting in a lower upper-bound for VL tracking. Besides, current VL trackers treat template and search branches as *homoplasmic* inputs, and adopt symmetrical feature learning structures for these two branches, inherited from typical vision-only Siamese tracking [20]. We argue the mixed modality may have different intrinsic nature than the pure vision modality, and thus requires a more flexible and general design for different signals.

**Our solution.** Having observed the above, we introduce a novel unified-adaptive vision-language representation, aiming for SOTA VL tracking *without using Transformer*[2]. Specifically, we first present modality mixer, or ModaMixer, a conceptually simple but effective module for VL interaction. Language is a high-level representation and its class embedding can help distinguish targets of different categories (*e.g.*, cat and dog) and meanwhile the attribute embedding (*e.g.*, color, shape) provides strong prior to separate targets of same class (*e.g.*, cars with different colors). The intuition is, channel features in vision representation also reveal semantics of objects [22, 56]. Inspired by this, ModaMixer regards language representation as a selector to reweight different channels of visual features, enhancing target-specific channels as well as suppressing irrelevant both intra- and inter-class channels. The selected feature is then fused with the original feature, using a special asymmetrical design (analyzed later in experiments), to generate the final unified VL representation. A set of ModaMixers are installed in a typical CNN from shallow to deep, boosting robustness and discriminability of the unified VL representation at different semantic levels. Despite simplicity, ModaMixer brings 6.9% gains over a pure CNN baseline [23] (*i.e.*, 50.7%→57.6%).

Despite huge improvement, the gap to SOTA Transformer tracker [8] remains (57.6% *v.s.* 64.9%). To mitigate the gap, we propose an asymmetrical searching strategy (ASS) to adapt the unified VL representation for improvements. Different from current VL tracking [20] adopting symmetrical and fixed template and search branches as in vision-only Siamese tracking [32], we argue that the learning framework of mixed modality should be adaptive and not fixed. To this end, ASS borrows the idea from neural architecture search (NAS) [63, 42] to separately learn *distinctive* and *asymmetrical* networks for mixed modality in different branches and ModaMixers. The asymmetrical architecture, to our best knowledge, is the first of its kind in matching-based tracking. Note, although NAS has been adopted in matching-based tracking [55], this method finds *symmetrical* networks for single modality. Differently, ASS is applied on mixed modality and the resulted architecture is *symmetrical*. Moreover, the network searched in ASS avoids burdensome re-training on ImageNet [14], enabling quick reproducibility of our work (only 0.625 GPU days with a single RTX-2080Ti). Our ASS is general and flexible, and together with ModaMixer, it surprisingly shows additional 7.6% gains (*i.e.*, 57.6%→65.2%), evidencing our argument and effectiveness of ASS.

Eventually, with the unified-adaptive representation, we implement the first pure CNN-based VL tracker that shows SOTA results comparable and even better than Transformer-based solutions, without bells and whistles. Specifically, we apply our method to a CNN baseline SiamCAR [23], and the resulted VL tracker VLT$_{\text{SCAR}}$ shows 65.2% SUC on LaSOT [17] while running at 43FPS, unbelievably improving the baseline by 14.5% and outperforming SOTA Transformer trackers [8, 52] (see again Fig. 1). We observe similar improvements by our approach on other four benchmarks. Besides empirical results, we provide theoretical analysis to evidence the effectiveness of our method. Note that, our approach is general in improving vision-only trackers including Transformer-based ones. We show this by applying it to TransT [8] and the resulted tracker VLT$_{\text{TT}}$ shows 2.4% SUC gains (*i.e.*, 64.9%→67.3%), evidencing its effectiveness and generality.

---

[2]Here we stress that we do not use Transformer for visual feature learning as in current Transformer trackers or for multimodal learning. We only use it in language embedding extraction (*i.e.*, BERT [15])

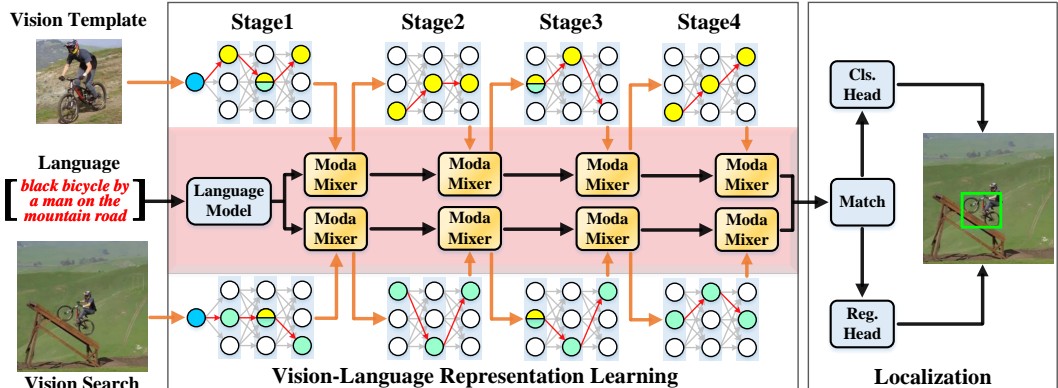

Figure 2: The proposed vision-language tracking framework. The semantic information of language description is injected to vision from shallow to deep layers of the asymmetrical modeling architecture to learn unified-adaptive vision-language representation.

We are aware that one can certainly leverage the Transformer [39] to learn a good (maybe better) VL representation for tracking, with larger data and longer training period. Different than this, our goal is to explore a cheaper way with simple architectures such as pure CNN for SOTA tracking performance and open more possibilities for future tracking beyond Transformer. In summary, our **contributions** are four-fold: **(i)** we introduce a novel unified-adaptive vision-language representation for SOTA VL tracking; **(ii)** we propose the embarrassingly simple yet effective ModaMixer for unified VL representation learning; **(iii)** we present ASS to adapt mixed VL representation for better tracking and **(iv)** using pure CNN architecture, we achieve SOTA results on multiple benchmarks.

## 2   Related Work

**Visual Tracking.** Tracking has witnessed great progress in the past decades. Particularly, Siamese tracking [5, 47], that aims to learn a generic matching function, is a representative branch and has revolutionized with numerous extensions [33, 61, 32, 57, 18, 25, 60, 62, 9]. Recently, Transformer [49] has been introduced to Siamese tracking for better interactions of visual features and greatly pushed the standard of state-of-the-art performance [8, 52, 46, 10, 35]. From a different perspective than using complex Transformer, we explore multimodal with simple CNN to achieve SOTA tracking.

**Vision-Language Tracking.** Natural language contains high-level semantics and has been leveraged to foster vision-related tasks [21, 31, 2] including tracking [34, 19, 20]. The work [34] first introduces linguistic description to tracking and shows that language enhances the robustness of vision-based method. Most recently, SNLT [20] integrates linguistic information into Siamese tracking by fusing results respectively obtained by vision and language. Different from these VL trackers that regard vision and language as independent cues with weak connections only at result fusion, we propose ModaMixer to unleash the power of VL tracking by learning unified VL representation.

**NAS for Tracking.** Neural architecture search (NAS) aims at finding the optimal design of deep network architectures [63, 42, 37, 26] and has been introduced to tracking [55, 60]. LightTrack [55] tends to search a lightweight backbone but is computationally demanding (about 40 V100 GPU days). AutoMatch uses DARTS [37] to find better matching networks for Siamese tracking. All these methods leverage NAS for vision-only tracking and search a *symmetrical* Siamese architecture. Differently, our work searches the network for multimodal tracking and tries to find a more general and flexible *asymmetrical* two-stream counterpart. In addition, our search pipeline only takes 0.625 RTX-2080Ti GPU days, which is much more resource-friendly.

## 3   Unified-Adaptive Vision-Language Tracking

This section details our unified-adaptive vision-language (VL) tracking as shown in Fig. 2. In specific, we first describe the proposed modality mixer for generating unified multimodal representation and then asymmetrical network which searches for learning adaptive VL representation. Afterwards, we illustrate the proposed tracking framework, followed by theoretical analysis of our method.

### 3.1 Modality Mixer for Unified Representation

The essence of multimodal learning is a simple and effective modality fusion module. As discussed before, existing VL trackers simply use a *later fusion* way, in which different modalities are treated independently and processed distantly until merging their final results [20, 34]. Despite the effectiveness to some extent, the complementarity of different modalities in representation learning is largely unexplored, which may impede the multimodal learning to unleash its power for VL tracking. In this work, we propose the modality mixer (dubbed **ModaMixer**) to demonstrate a compact way to learn a unified vision-language representation for tracking.

ModaMixer considers language representation as selector to reweight channels of vision features. In specific, given the language description with $N$ words of a video[3], a language model [15] is adopted to abstract the sentence to semantic features with size of $(N+2) \times d$. The extra "2" denotes the "[CLS][SEP]" characters in language model processing (see [15] for more details). Notably, descriptions for different videos may contain various length $N$. To ensure the ModaMixer applicable for all videos, we first average the features for all words along sequence length dimension "(N+2)" to generate

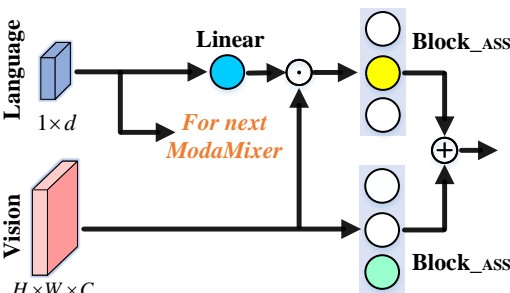

Figure 3: Illustration of the ModaMixer.

a unique language representation $\mathbf{f}_l \in \mathbb{R}^{1 \times d}$ for each description. Then a linear layer is followed to align the channel number of $\mathbf{f}_l$ with the corresponding vision feature $\mathbf{f}_v \in \mathbb{R}^{H \times W \times C}$. Channel selector is expressed as Hadamard product operator, which point-wisely multiplies language representation $(1 \times C)$ to embedding of each spatial position in the vision feature $\mathbf{f}_v$. Finally, a residual connection between the mixed feature $\mathbf{f}_m$ and vision feature $\mathbf{f}_v$ is conducted to avoid losing informative vision details. In a nutshell, the ModaMixer can be formulated as,

$$\mathbf{f}_m = \text{Block}_{\text{ASS}}\left(\text{Linear}(\mathbf{f}_l) \odot \mathbf{f}_v\right) + \text{Block}_{\text{ASS}}\left(\mathbf{f}_v\right), \tag{1}$$

where $\odot$ denotes Hadamard product, $\text{Linear}$ is a linear projection layer with weight matrix size of $d \times C$ for channel number alignment, and $\text{Block}_{\text{ASS}}$ indicates post-processing block before residual connection. Please note that, to enable adaptive feature modeling for different modalities, we search different $\text{Block}_{\text{ASS}}$ to process features before and after fusion (see Sec. 3.2 for more details). The proposed ModaMixer is illustrated in Fig. 3. Akin to channel attention [44, 22], the high-level semantics in language representation dynamically enhance target-specific channels in vision features, and meanwhile suppress the responses of distractors belonging to both inter- and intra-classes.

### 3.2 Asymmetrical Search for Adaptive Vision-Language Representation

Besides the fusion module, the other crucial key for vision-language tracking is how to construct the basic modeling structure. The simplest strategy is to inherit a symmetrical Siamese network from vision-based tracking (*e.g.*, [5, 32]), as in current VL trackers [20]. But the performance gap still remains if using this manner, which is mostly blamed on the neglect of the different intrinsic nature between VL-based multimodal and vision-only single modality. To remedy this, we propose an asymmetrical searching strategy (dubbed **ASS**) to learn an adaptive modeling structure for pairing with ModaMixer.

The spirits of network search are originated from the field of Neural Architecture Search (NAS). We adopt a popular NAS model, in particular the single-path one-shot method SPOS [26], for searching the optimal structure of our purpose. Although SPOS has been utilized for tracking [55], our work significantly differs from it from two aspects: **1)** Our ASS is tailored for constructing an *asymmetrical* two-stream network for *multimodal* tracking, while [55] is designed to find a *symmetrical* Siamese network for vision-only *single-modality* tracking. Besides, we search layers both in the backbone network and the post-processing $\text{Block}_{\text{ASS}}$ in the ModaMixer (see Eq. 1); **2)** Our ASS reuses the

---

[3]The language description in tracking is generated *only* by the initial target object in the first frame.

Table 1: The asymmetrical architecture learned by ASS. 🔵 is the stem convolution layer. ( 🔴🟠🟢🔵 ) represents the basic ASS unit, where the first three ones indicate *Shuffle block* [59] with kernel sizes of (3,5,7), respectively, and the last one denotes a *Shuffle Xception block* [59] with kernel size of 3.

| | Stem | Stage1 | Moda Mixer | Stage2 | Moda Mixer | Stage3 | Moda Mixer | Stage4 | Moda Mixer |
|---|---|---|---|---|---|---|---|---|---|
| **Template** | 🔵 | 🟢🟢🟢 | 🔴🔴 | 🔵🔵🔵 | 🔵🔴 | 🔴🟢🟢 / 🔵🟢🟢🟢 | 🔵🔵 | 🟢🟢🟢 | 🟠🔴 |
| **Search** | 🔵 | 🟠🔵🔴 | 🟢🟢 | 🟠🟢🟢 | 🔵🔴 | 🟢🟢🟢 / 🔴🔵🟢🟢 | 🔵🔵 | 🔵🔵🔵 | 🔴🔴 |

pre-trained supernet from SPOS, which avoids the burdensome re-training on ImageNet [14] (both for the supernet and found subnet) and thus reduces the time complexity of our search pipeline to $1/64$ of that in LightTrack [55] (*i.e.*, **0.625 RTX-2080Ti GPU days** *v.s.* **40 V100 GPU days**). Due to limited space, please refer to **appendix** for more details and comparison of our ASS and [55].

The search space and search strategy of ASS are kept consistent with the original SPOS [26]. In particular, the search pipeline is formulated as,

$$W_{\mathcal{A}} = \underset{W}{\operatorname{argmin}} \ \mathbb{E}_{a \sim \Gamma(\mathcal{A})} \left[ \mathcal{L}_{\text{train}}(\mathcal{N}(a, W(a))) \right], \tag{2}$$

$$a^* = \underset{a \in \mathcal{A}}{\operatorname{argmax}} \ \text{SUC}_{\text{val}} \left( \mathcal{N}(a, W_{\mathcal{A}}(a)) \right), \tag{3}$$

where $\mathcal{A}$ represents the architecture search space of the network $\mathcal{N}$, $a$ is a sample from $\mathcal{A}$ and $W$ denotes the corresponding network weights. Notably, the network $\mathcal{N}$ includes three components $\mathcal{N} = \{\varphi_t, \varphi_s, \varphi_m\}$, where each indicates backbone for the template branch $\varphi_t$, backbone for the search branch $\varphi_s$ and layers in the ModaMixer $\varphi_m$. The whole pipeline consists of training supernet on tracking datasets via random sampling $\Gamma$ from search space $\mathcal{A}$ (Eq. 2) and finding the optimal subnet via evolutionary algorithms (Eq. 3). The SUC (success score) on validation data is used as rewards of evolutionary algorithms. Tab. 1 demonstrates the searched asymmetrical networks in our VL tracking. For more details of ASS, we kindly refer readers to **appendix** or [26].

### 3.3 Tracking Framework

With the proposed ModaMixer and the searched asymmetrical networks, we construct a new vision-language tracking framework, as shown in Fig. 2 and Tab. 1. Our framework is matching-based tracking. Both template and search backbone networks contain 4 stages with the maximum stride of 8, the chosen blocks of each stage are denoted with different colors in Tab. 1. ModaMixer is integrated into each stage of the template and search networks to learn informative mixed representation. It is worth noting that, the asymmetry is revealed in not only the design of backbone networks, but also the ModaMixer. Each ModaMixer shares the same meta-structure as in Fig. 3, but comprises different post-processing layers $\text{Block}_{\text{ASS}}$ to allow adaption to different semantic levels (*i.e.*, network depth) and input signals (*i.e.*, template and search, pure-vision and mixed feature in each ModaMixer). With the learned unified-adaptive VL representations from the template and search branches, we perform feature matching and target localization, the same as in our baseline.

### 3.4 A Theoretical Explanation

This section presents a theoretical explanation of our method, following the analysis in [29]. Based on the Empirical Risk Minimization (ERM) principle [45], the objective of representation learning is to find better network parameters $\theta$ by minimizing the empirical risk, *i.e.*,

$$\min \ \hat{r}(\theta_{\mathcal{M}}) \triangleq \frac{1}{n} \sum_{i=1}^{n} \mathcal{L}(\mathcal{X}_i, y_i; \theta_{\mathcal{M}}) \quad \text{s.t.} \ \theta_{\mathcal{M}} \in \mathcal{F}. \tag{4}$$

where $\mathcal{L}$ denotes loss function, $\mathcal{M}$ represents the modality set, $n$ indicates sample number, $\mathcal{X}_i = \{x_i^1, x_i^2 ... x_i^{|\mathcal{M}|}\}$ is the input mutimodal signal, $y_i$ is training label, and $\mathcal{F}$ demotes optimization space

Table 2: State-of-the-art comparisons on LaSOT [17], LaSOT_Ext [16], TNL2K [53], GOT-10k [28] and OTB99-LANG (OTB99-L) [34]. TransT and SiamCAR are baselines of the proposed **VLT**$_{\text{TT}}$ and **VLT**$_{\text{SCAR}}$, respectively. $^0$ and $^t$ denote the settings of "0-tensor" and "template" without language description. All metrics of performance are in % in tables unless otherwise specified.

| Type | Method | LaSOT | | LaSOT$_{\text{Ext}}$ | | TNL2K | | GOT-10k | | | OTB99-L | |
|------|--------|-------|---|---------|---|-------|---|---------|---|---|---------|---|
| | | SUC | P | SUC | P | SUC | P | AO | SR$_{0.5}$ | SR$_{0.75}$ | SUC | P |
| CNN-based | SiamRCNN [50] | 64.8 | 68.4 | - | - | 52.3 | 52.8 | 64.9 | 72.8 | 59.7 | 70.0 | 89.4 |
| | PrDiMP [13] | 59.8 | 60.8 | - | - | 47.0 | 45.9 | 63.4 | 73.8 | 54.3 | 69.5 | 89.5 |
| | AutoMatch [60] | 58.3 | 59.9 | 37,6 | 43.0 | 47.2 | 43.5 | 65.2 | 76.6 | 54.3 | 71.6 | 93.2 |
| | Ocean [62] | 56.0 | 56.6 | - | - | 38.4 | 37.7 | 61.1 | 72.1 | 47.3 | 68.0 | 92.1 |
| | KYS [6] | 55.4 | - | - | - | 44.9 | 43.5 | 63.6 | 75.1 | 51.5 | - | - |
| | ATOM [12] | 51.5 | 50.5 | 37.6 | 43.0 | 40.1 | 39.2 | 55.6 | 63.4 | 40.2 | 67.6 | 82.4 |
| | SiamRPN++ [32] | 49.6 | 49.1 | 34.0 | 39.6 | 41.3 | 41.2 | 51.7 | 61.6 | 32.5 | 63.8 | 82.6 |
| | C-RPN [18] | 45.5 | 42.5 | 27.5 | 32.0 | - | - | - | - | - | - | - |
| | SiamFC [5] | 33.6 | 33.9 | 23.0 | 26.9 | 29.5 | 28.6 | 34.8 | 35.3 | 9.8 | 58.7 | 79.2 |
| | ECO [11] | 32.4 | 30.1 | 22.0 | 24.0 | - | - | 31.6 | 30.9 | 11.1 | - | - |
| | SiamCAR [23] | 50.7 | 51.0 | 33.9 | 41.0 | 35.3 | 38.4 | 56.9 | 67.0 | 41.5 | 68.8 | 89.1 |
| CNN-VL | SNLT [20] | 54.0 | 57.6 | 26.2 | 30.0 | 27.6 | 41.9 | 43.3 | 50.6 | 22.1 | 66.6 | 80.4 |
| | **VLT**$_{\text{SCAR}}^{0}$ (Ours) | **65.2** | **69.1** | **41.2** | **47.5** | **48.3** | **46.6** | **61.4** | **72.4** | **52.3** | **72.7** | **88.8** |
| | **VLT**$_{\text{SCAR}}^{t}$ (Ours) | **63.9** | **67.9** | **44.7** | **51.6** | **49.8** | **51.1** | **61.0** | **70.8** | **52.2** | **73.9** | **89.8** |
| Trans-based | STARK [54] | 66.4 | 71.2 | 47.8 | 55.1 | - | - | 68.0 | 77.7 | 62.3 | 69.6 | 91.4 |
| | TrDiMP [52] | 63.9 | 66.3 | - | - | - | - | 67.1 | 77.7 | 58.3 | 70.5 | 92.5 |
| | TransT [8] | 64.9 | 69.0 | 44.8 | 52.5 | 50.7 | 51.7 | 67.1 | 76.8 | 60.9 | 70.8 | 91.2 |
| Trans-VL | **VLT**$_{\text{TT}}^{0}$ (Ours) | **66.3** | **70.5** | **45.4** | **52.1** | **52.2** | **52.1** | **68.4** | **81.5** | **62.4** | **74.7** | **91.2** |
| | **VLT**$_{\text{TT}}^{t}$ (Ours) | **67.3** | **72.1** | **48.4** | **55.9** | **53.1** | **53.3** | **69.4** | **81.1** | **64.5** | **76.4** | **93.1** |

of $\theta$. Given the empirical risk $\hat{r}(\theta_{\mathcal{M}})$, its corresponding population risk is defined as,

$$r(\theta_{\mathcal{M}}) = \mathbb{E}_{(\mathcal{X}_i, y_i) \sim \mathcal{D}_{train}} [\hat{r}(\theta_{\mathcal{M}})] \tag{5}$$

Following [1, 48, 29], the population risk is adopted to measure the learning quality. Then the **latent representation quality** [29] is defined as,

$$\eta(\theta) = \inf_{\theta \in \mathcal{F}} [r(\theta) - r(\theta^*)] \tag{6}$$

where $*$ represents the optimal case, inf indicates the best achievable population risk. With the empirical Rademacher complexity $\mathfrak{R}$ [4], we restate the conclusion in [29] with our definition.

**Theorem 1** ([29]). *Assuming we have produced the empirical risk minimizers $\hat{\theta}_{\mathcal{M}}$ and $\hat{\theta}_{\mathcal{S}}$, training with the $|\mathcal{M}|$ and $|\mathcal{S}|$ modalities separately ($|\mathcal{M}| > |\mathcal{S}|$). Then, for all $1 > \delta > 0$, with probability at least $1 - \frac{\delta}{2}$:*

$$r\left(\hat{\theta}_{\mathcal{M}}\right) - r\left(\hat{\theta}_{\mathcal{S}}\right) \leq \gamma_{\mathcal{D}}(\mathcal{M}, \mathcal{S}) + 8L\mathfrak{R}_n(\mathcal{F}_{\mathcal{M}}) + \frac{4C}{\sqrt{n}} + 2C\sqrt{\frac{2\ln(2/\delta)}{n}} \tag{7}$$

*where*

$$\gamma_{\mathcal{D}}(\mathcal{M}, \mathcal{S}) \triangleq \eta(\hat{\theta}_{\mathcal{M}}) - \eta(\hat{\theta}_{\mathcal{S}}) \qquad \mathfrak{R}_n(\mathcal{F}_{\mathcal{M}}) \sim \sqrt{Complexity(\mathcal{F}_{\mathcal{M}})/n} \tag{8}$$

$\gamma_{\mathcal{D}}(\mathcal{M}, \mathcal{S})$ computes the quality difference learned from multiple modalities $\mathcal{M}$ and single modality $\mathcal{S}$ with dataset $\mathcal{D}$. Theorem 1 defines an upper bound of the population risk training with different number of modalities, **which proves that more modalities could potentially enhance the representation quality.** Furthermore, the Rademacher complexity $\mathfrak{R}_n(\mathcal{F})$ is proportional to the network complexity, **which demonstrates that heterogeneous network would theoretically rise the upper bound of** $r\left(\hat{\theta}_{\mathcal{M}}\right) - r\left(\hat{\theta}_{\mathcal{S}}\right)$**, and also exhibits that our asymmetrical design has larger optimization space when learning with $|\mathcal{M}|$ modalities compared to $|\mathcal{S}|$ modalities ($|\mathcal{M}| > |\mathcal{S}|$).** The proof is beyond our scoop, and please refer to [29] for details.

# 4 Experiment

## 4.1 Implementation Details

We apply our method to both CNN-based SiamCAR [23] (dubbed **VLT**$_{\text{SCAR}}$) and Transformer-based TransT [8] (dubbed **VLT**$_{\text{TT}}$). The matching module and localization head are inherited from the baseline tracker without any modifications.

**Searching for VLT.** The proposed ASS aims to find a more flexible modeling structure for vision-language tracking (VLT). Taking VLT$_{\text{SCAR}}$ as example, the supernet from SPOS [26] is used as feature extractor to replace the ResNet [27] in SiamCAR. We train the trackers with supernet using training splits of COCO [36], Imagenet-VID [14], Imagenet-DET [14], Youtube-BB [43], GOT-10k [28], LaSOT [17] and TNL2K [53] for 5 epochs, where each epoch contains $1.2 \times 10^6$ template-search pairs. Once finishing supernet training, evolutionary algorithms as in SPOS [26] is applied to search for optimal subnet and finally obtains VLT$_{\text{SCAR}}$. The whole search pipeline consumes 15 hours on a single RTX-2080Ti GPU. The search process of VLT$_{\text{TT}}$ is similar to VLT$_{\text{SCAR}}$. We present more details in the **appendix** due to space limitation.

**Optimizing VLT**$_{\text{SCAR}}$ **and VLT**$_{\text{TT}}$**.** The training protocol of VLT$_{\text{SCAR}}$ and VLT$_{\text{TT}}$ follows the corresponding baselines SiamCAR [23] and TransT [8]. Notably, for each epoch, half training pairs come from datasets without language annotations (*i.e.*, COCO [36], Imagenet-VID [14], Imagenet-DET [14], Youtube-BB [43]). The language representation is set as 0-tensor or visual pooling feature under this circumstances (discussed in Sec. 4.4)[4].

## 4.2 State-of-the-art Comparison

Tab. 2 presents the results and comparisons of our trackers with other SOTAs on LaSOT [17], LaSOT$_{\text{Ext}}$ [16], TNL2K [53], OTB99-LANG [34] and GOT-10K [28]. The proposed VLT$_{\text{SCAR}}$ and VLT$_{\text{TT}}$ run at 43/35 FPS on a single RTX-2080Ti GPU, respectively. Compared with the speeds of baseline trackers SiamCAR [23]/TransT [8] with 52/32 FPS, the computation cost of our method is small. Moreover, our VLT$_{\text{TT}}$ outperforms TransT in terms of both accuracy and speed.

Compared with SiamCAR [23], VLT$_{\text{SCAR}}$ achieves considerable SUC gains of 14.5%/10.8%/14.5% on LaSOT/LaSOT$_{\text{Ext}}$/TNL2K, respectively, which demonstrates the effectiveness of the proposed VL tracker. Notably, our VLT$_{\text{SCAR}}$ outperforms the current best VL tracker SNLT [20] for 11.2%/18.5% on LaSOT/LaSOT$_{\text{Ext}}$, showing that the unified-adaptive vision-language representation is more robust for VL tracking and is superior to simply fusing tracking results of different modalities. The advancement of our method is preserved across different benchmarks. What surprises us more is that the CNN-based VLT$_{\text{SCAR}}$ is competitive and even better than recent vision Transformer-based approaches. For example, VLT$_{\text{SCAR}}$ outperforms TransT [8] on LaSOT and meanwhile runs faster (43 FPS *v.s.* 32 FPS) and requires less training pairs ($2.4 \times 10^7$ *v.s.* $3.8 \times 10^7$). By applying our method to TransT, the new tracker VLT$_{\text{TT}}$ improves the baseline to 67.3% in SUC with 2.4% gains on LaSOT while being faster, showing its generality.

## 4.3 Component-wise Ablation

We analyze the influence of each component in our method to show the effectiveness and rationality of the proposed ModaMixer and ASS. The ablation experiments are conducted on VLT$_{\text{SCAR}}$ with "0-tensor" setting (discussed in Sec. 4.4), and results are presented in Tab. 3. By directly applying the ModaMixer on the baseline SiamCAR [23] ("ResNet50+ModaMixer"), it obtains SUC gains of 6.9% on LaSOT (②*v.s.*①). This verifies that the unified VL representation effectively improves tracking robustness. One interesting observation is that ASS improves vision-only baseline for 1.4% percents on LaSOT (③*v.s.*①), but when equipping with ModaMixer, it surprisingly further brings 7.6% SUC gains (④*v.s.*②), which shows the complementary of multimodal representation learning (ModaMixer) and the proposed ASS.

---

[4]GOT-10k [28] provides simple descriptions for object/motion/major/root class, *e.g.*, "dove, walking, bird, animal", in each video. We concatenate these words to obtain a pseudo language description.

Table 3: Ablation on ModaMixer and asymmetrical searching strategy (ASS).

| # | Method | ModaMixer | ASS | LaSOT | | | TNL2K | | |
|---|--------|-----------|-----|-------|---|---|-------|---|---|
| | | | | **SUC** | $\mathbf{P}_{\text{Norm}}$ | **P** | **SUC** | $\mathbf{P}_{\text{Norm}}$ | **P** |
| ① | Baseline | - | - | 50.7 | 60.0 | 51.0 | 35.3 | 43.6 | 38.4 |
| ② | **VLT**$_{\text{SCAR}}$ | $\checkmark$ | - | 57.6 | 65.8 | 61.1 | 41.5 | 49.2 | 43.2 |
| ③ | **VLT**$_{\text{SCAR}}$ | - | $\checkmark$ | 52.1 | 59.8 | 50.6 | 40.7 | 47.2 | 40.2 |
| ④ | **VLT**$_{\text{SCAR}}$ | $\checkmark$ | $\checkmark$ | **65.2** | **74.9** | **69.1** | **48.3** | **55.2** | **46.6** |

Table 4: Comparisons with two strategies (*i.e.,* "0-tensor" and "template") and different language settings during inference.

| Method | Settings | | LaSOT | | | TNL2K | | |
|--------|----------|--|-------|---|---|-------|---|---|
| | | | **SUC** | $\mathbf{P}_{\text{Norm}}$ | **P** | **SUC** | $\mathbf{P}_{\text{Norm}}$ | **P** |
| **VLT**$_{\text{SCAR}}$ | **0-tensor** | **w/. language** | 65.2 | 74.9 | 69.1 | 48.3 | 55.2 | 46.6 |
| | | **w/o. language** | 50.8 | 57.9 | 52.6 | 39.5 | 47.1 | 41.2 |
| | | **Pse. language** | 53.1 | 60.4 | 55.0 | 38.1 | 45.7 | 39.6 |
| | **template** | **w/. language** | 63.9 | 73.3 | 67.9 | 49.8 | 58.3 | 51.1 |
| | | **w/o. language** | 53.4 | 60.7 | 54.6 | 41.1 | 49.1 | 42.9 |
| | | **Pse. language** | 51.6 | 58.1 | 53.4 | 38.8 | 46.6 | 40.5 |
| **VLT**$_{\text{TT}}$ | **0-tensor** | **w/. language** | 66.3 | 77.0 | 70.5 | 52.2 | 58.6 | 52.1 |
| | | **w/o. language** | 60.7 | 71.1 | 63.1 | 48.2 | 54.6 | 46.8 |
| | | **Pse. language** | 59.3 | 68.6 | 62.2 | 49.3 | 55.7 | 49.2 |
| | **template** | **w/. language** | 67.3 | 78.0 | 72.1 | 53.1 | 59.3 | 53.3 |
| | | **w/o. language** | 61.0 | 71.5 | 63.4 | 49.1 | 55.5 | 48.3 |
| | | **Pse. language** | 59.7 | 69.2 | 63.0 | 50.0 | 56.3 | 50.3 |

## 4.4 Further Analysis

**Dealing with videos without language description during training.** As mentioned above, language annotations are not provided in several training datasets (*e.g.,* YTB-BBox [43]). We design two strategies to handle that. One is to use "0-tensor" as language embedding, and the other is to replace the language embedding with visual features which are generated by pooling template feature in the bounding box (dubbed as "template"). As shown in Tab. 2, the two strategies perform competitively, but the one with visual feature (*i.e.,* VLT$_{\text{SCAR}}^{t}$ and VLT$_{\text{TT}}^{t}$) is slightly better in average.

**No/Pseudo description during inference.** VL trackers require the first frame of a video is annotated with a language description. One may wonder that what if there is no language description? Tab. 4 presents the results by removing the description and using that generated with an recent advanced image-caption method [41] (in ICML2021) based on VLT$_{\text{SCAR}}$ and VLT$_{\text{TT}}$ with "0-tensor" and "template" settings. The results show that, without language description, tracking performances heavily degrade (*e.g.,* 63.9% → 53.4%, 67.3% → 61.0% SUC on LaSOT of VLT$_{\text{SCAR}}^{t}$ and VLT$_{\text{TT}}^{t}$, respectively), verifying that the high-level semantics in language do help in improving robustness. Even though, the performances are still better than the vision-only baseline. Surprisingly, when using the generated description, it doesn't show promising results (*e.g.,* 51.6% of VLT$_{\text{SCAR}}^{t}$ and 59.7% of VLT$_{\text{TT}}^{t}$), indicating that it is still challenging to generate accurate caption in real-world cases and noisy caption even brings negative effects to the model.

**Symmetrical or Asymmetrical?** The proposed asymmetrical searching strategy is essential for achieving an adaptive vision-language representation. As illustrated in Tab. 5a, we experiment by searching for a symmetrical network (including both backbone and Block$_{\text{ASS}}$ in the ModaMixer) based on VLT$_{\text{SCAR}}^{t}$, but it is inferior to the asymmetrical counterpart for 3.9%/6.2% of success rate (SUC) and precision (P) on LaSOT [17], respectively, which empirically proves our argument.

**Asymmetry in ModaMixer.** The asymmetry is used in not only the backbone network, but also the ModaMixer. In our work, the post-processing layers for different signals (visual and mixed features) are decided by ASS, which enables the adaption at both semantic levels (*i.e.*, network depth) and different input signals (*i.e.*, template and search, pure-vision and mixed feature in each

Table 5: Evaluating different settings on LaSOT: (a) the influence of symmetrical and our asymmetrical design, (b) adopting fixed ShuffleNet block or searching the post-processing block in ModaMixer, and (c) removing the residual connection (dubbed as "Res") of ModaMixer.

(a)

| Settings | SUC | P |
|---|---|---|
| **symmetrical** | 60.0 | 61.7 |
| **asymmetrical** | 63.9 | 67.9 |

(b)

| Settings | SUC | P |
|---|---|---|
| **Shuffle-ModaMixer** | 59.1 | 62.2 |
| **NAS-ModaMixer** | 63.9 | 67.9 |

(c)

| Settings | SUC | P |
|---|---|---|
| **w/o. Res** | 61.1 | 63.6 |
| **w/. Res** | 63.9 | 67.9 |

Table 6: Comparing different data volumes and sources for training. "SiamCAR Four Datasets" consist of VID, YOUTUBEBB, DET and COCO, "SiamCAR Seven Datasets" consist of VID, YOUTUBEBB, DET, COCO, GOT-10K, LaSOT and TNL2K, "TransT Four Datasets" consist of COCO, GOT-10K, LaSOT and TrackingNet.

| # | Method | Data Volume | Data Source | LaSOT | | TNL2K | |
|---|---|---|---|---|---|---|---|
| | | | | SUC | P | SUC | P |
| ① | SiamCAR | 60W×20Epoch | SiamCAR Four Datasets | 50.7 | 51.0 | 35.3 | 38.4 |
| ② | SiamCAR | 60W×20Epoch | LaSOT | 51.6 | 52.3 | 35.0 | 36.4 |
| ③ | SiamCAR | 120W×20Epoch | SiamCAR Seven Datasets | 48.7 | 46.6 | 39.7 | 39.2 |
| ④ | $\text{VLT}_{\text{SCAR}}$ | 60W×20Epoch | LaSOT | 57.0 | 58.6 | 39.0 | 39.8 |
| ⑤ | $\text{VLT}_{\text{SCAR}}$ | 120W×20Epoch | SiamCAR Seven Datasets | 63.9 | 67.9 | 48.3 | 46.6 |
| ⑥ | TransT | 3.8W×1000Epoch | TransT Four Datasets | 64.9 | 69.0 | 50.7 | 51.7 |
| ⑦ | TransT | 3.8W×1000Epoch | TransT Four Datasets, TNL2K | 62.2 | 65.2 | 51.2 | 52.3 |
| ⑧ | $\text{VLT}_{\text{TT}}$ | 3.8W×1000Epoch | TransT Four Datasets, TNL2K | 67.3 | 72.1 | 53.1 | 53.3 |

ModaMixer). As in Tab. 5b, when replacing the post-processing layers with a fixed ShuffleNet block from SPOS [26] (*i.e.*, inheriting structure and weights from the last block in each backbone stage), the performance of $\text{VLT}_{\text{SCAR}}^t$ drops from $63.9\%$ to $59.1\%$ in SUC on LaSOT. This reveals that the proposed ASS is important for building a better VL learner.

**Residual Connection of ModaMixer** The residual connection [27] is a commonly used trick to avoid information loss. In our VL representation learning, it provides more vision messages for better multimodal fusion. We experiment by removing the structure based on $\text{VLT}_{\text{SCAR}}$ with the "template" setting as shown in Tab. 5c. Compared to the default setting (*i.e., w/*. Res), the loss of additional vision details brings decreases for $2.7\%/2.1\%$ of SUC on LaSOT/TNL2K, respectively. Even though, the performance is still much higher than the baseline. This demonstrates the improvements are mainly attributed to the multimodal fusion.

**Volume and Source of Training Data.** As common wisdom, the tracking performance is deeply influenced by training data volume and source. As illustrated in Tab. 6, we experiment by comparing $\text{VLT}_{\text{SCAR}}^t$, $\text{VLT}_{\text{TT}}^t$ and their baselines (*i.e.*, SiamCAR and TransT) with different data settings:

(1) We retrain SiamCAR with the same data setting of $\text{VLT}_{\text{SCAR}}$ (③ *v.s.* ⑤). Compared to the default setting (①), double data volume and three more data sources contain different biases, which affect the trained model to produce biased outcomes, as illustrated in [7, 30, 38]. From the results (③), the addition of TNL2K significantly improves the default ① with $4.4\%$ gains in SUC on TNL2K, whereas the performance on LaSOT slightly decreases. Compared to $\text{VLT}_{\text{SCAR}}$ with the same setting (⑤), ③ is still suppressed for $8.6\%/7.4\%$ of SUC and P on TNL2K, respectively.

(2) We also retrain $\text{VLT}_{\text{SCAR}}$ with the only LaSOT (④), which keeps aligned with ②. The SUC scores on LaSOT and TNL2K degrade heavily to $57.0\%/39.0\%$ compared to the default ⑤, respectively. **This is caused by the great reduction of the language-annotated training data, i.e., from 1120 (LaSOT)+1300 (TNL2K)+9335 (GOT-10k) to 1120 (LaSOT).** Our model is hard to learn a good multimodal representation with the quite less language-annotated training data, which violates our intention. Even though, our $\text{VLT}_{\text{SCAR}}$ (④) still outperforms the baseline SiamCAR (②) for $5.4\%/4.0\%$ of SUC scores on LaSOT/TNL2K.

(3) TransT is also retrained with the same data setting as $\text{VLT}_{\text{TT}}$ (⑦ *v.s.* ⑧). More data sources bring similar biases and influence the performance as SiamCAR, compared to default TransT (⑥). Our $\text{VLT}_{\text{TT}}$ (⑧) still achieves superior scores on both LaSOT and TNL2K.

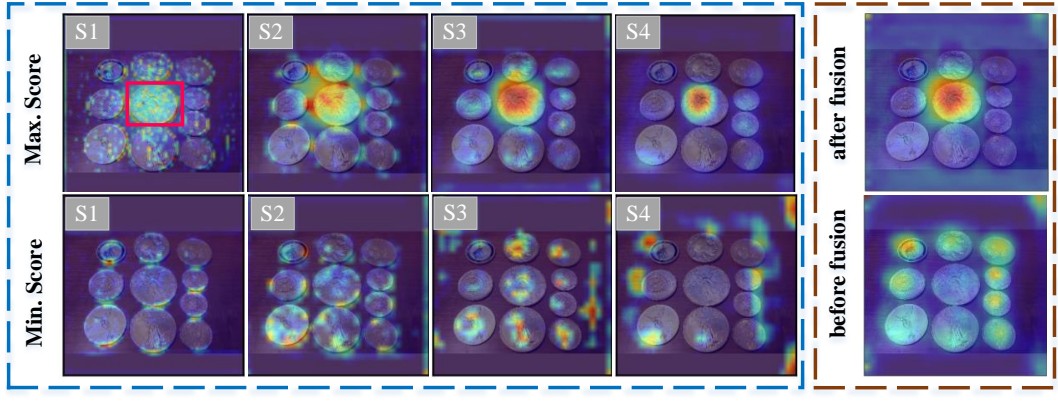

**(a) Channels visualization with max/min selection scores**          **(b) Activation maps**

Figure 4: (a) feature channel with maximum/minimum (top/bottom) selection scores from ModaMixer in stage1-4. (b) activation map before/after (top/bottom) multimodal fusion in ModaMixer.

Table 7: Results of applying ModaMixer and ASS to SiamRPN++.

| Method | LaSOT | | | TNL2K | | |
|---|---|---|---|---|---|---|
| | **SUC** | $\mathbf{P}_{\text{Norm}}$ | **P** | **SUC** | $\mathbf{P}_{\text{Norm}}$ | **P** |
| **VLT**$_{\text{RPN++}}$ | 59.0 | 68.4 | 62.6 | 45.8 | 54.2 | 47.4 |
| **SiamRPN++** | 49.6 | 56.9 | 49.1 | 41.3 | 48.2 | 41.2 |

**Channel selection by ModaMixer.** ModaMixer translates the language description to a channel selector to reweight visual features. As shown in Fig. 4, the channel activation maps with maximum selection scores always correspond to the target, while the surrounding distractors are successfully assigned with minimum scores (Fig. 4 (a)-bottom). Besides, with multimodal fusion (or channel selection), the network can enhance the response of target and meanwhile suppress the distractors (see Fig. 4 (b)). This evidences our argument that language embedding can identify semantics in visual feature channels and effectively select useful information for localizing targets. More visualization results are presented in **appendix** due to limited space.

**Multimodal Vision-Language Tracking with SiamRPN++ [32]** We apply our method to another pure CNN-based tracker SiamRPN++ (dubbed VLT$_{\text{RPN++}}$) and the results are shown in Tab. 7. Compared with the baseline SiamRPN++, VLT$_{\text{RPN++}}$ achieves considerable SUC gains of $9.4\%/4.5\%$ on LaSOT/TNL2K, respectively. This demonstrates the effectiveness of multimodal representation learning (ModaMixer) and the proposed ASS.

## 5   Conclusion

In this work, we explore a different path to achieve SOTA tracking without complex Transformer, *i.e.,* multimodal VL tracking. The essence is a unified-adaptive VL representation, learned by our ModaMixer and asymmetrical networks. In experiments, our approach surprisingly boosts a pure CNN-based Siamese tracker to achieve competitive or even better performances compared to recent SOTAs. Besides, we provide an theoretical explanation to evidence the effectiveness of our method. We hope that this work inspires more possibilities for future tracking beyond Transformer.

## Acknowledgments

This work was supported by the National Natural Science Foundation of China under Grant (62176020), the National Key Research and Development Program (2020AAA0106800), the Beijing Natural Science Foundation under Grant (Z180006, L211016), CAAI-Huawei MindSpore Open Fund and Chinese Academy of Sciences (OEIP-O-202004).

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
