# OpenReview forum: "Divert More Attention to Vision-Language Tracking"
_NeurIPS.cc/2022/Conference — NeurIPS 2022 Accept_

### Official Review · Reviewer_YvUr · 2022-07-02

**Rating:** 5
**Confidence:** 5
**Soundness:** 2 fair
**Presentation:** 3 good
**Contribution:** 2 fair

**Summary:**

This paper proposes a method to learn the vision-language representation for achieving SOTA tracking by using pure ConvNets. It proposes a modality mixer as the fusion module and uses it in both low-level ConvStages and high-level ConvStages to explore the complementarity of different modalities. It also uses NAS to search an asymmetrical siamese network to learn adaptive vision-language representations. The proposed method is also used to improve a Transformer-based tracker.

**Questions:**

-The proposed trackers and the baselines should be trained with the same training set.
-More quantitative studies about the modality mixer should be supplemented.
-During inference, using the 0-tensor or the visual pooling feature for further analysis.
-See weaknesses.

**Ethics Review Area:**

["I don’t know"]

**Strengths And Weaknesses:**

Strengths:
+The proposed method can make a CNN-based tracker and a Transformer-based tracker achieve SOTA performance on five benchmarks. The performance of vision-language tracking is improved largely.

+This work uses NAS for tracking to search different model structures for template and search region respectively. This is a novel practice for siamese trackers.

 Weaknesses:
-The proposed trackers and the baselines use different training data. The comparisons in this paper are unfair. For instance, both SiamCAR and TransT didn’t use TNL2K for model training, they shouldn’t be used as the baselines for comparison on TNL2K. SiamCAR only uses the LaSOT training set for model learning to achieve 50.7SUC. But the proposed trackers use lots of extra training sets. So the performance gains shown in Figure1 are debatable, and the current ablation study cannot show the effectiveness of the proposed method.

-Modality Mixer is too simple to be the main contribution. It’s cross-modal channel attention. The authors attempted to make the structure more complex by designing a residual connection. But this modification brings some doubts about the effectiveness of the cross-modal channel attention. We don’t know whether the final improvements can be attributed to the modal-fusion or the pure vision feature. This paper didn’t provide any ablation analysis about it. The current experiments are not sufficient.

-This paper proposes a VL representation learning method for object tracking. But half training pairs don’t have a language description, and the authors use the 0-tensor or the visual feature as the pseudo language description for training. It's doubtful that the learned representations are really multi-modal representations. The authors conducted an experiment by removing the description to prove the effectiveness of modal fusion. But how about using a 0-tensor or a visual pooling feature for inference? The current experiments are insufficient.

---

> ### Author Response · Authors · 2022-08-01
> **Response to Reviewer YvUr (part 3)**
>
> ***Q3: Ablation experiments of using a 0-tensor or a visual pooling feature for inference are required.***
>
> **A3**: Thanks for this constructive comment. We agree that, in order to validate the effectiveness of modal fusion, it is crucial to conduct the ablation experiments of using a 0-tensor or a visual pooling feature for inference (as discussed in Tab. 5 (c) of the manuscript). As suggested by the reviewer, we conduct such an ablation as shown in Tab. #3. From Tab. #3, we can see that, when removing language from the tracking inference, the performance of VLT_SCAR heavily drops from 65.2%/48.3% to 50.8%/39.5% in SUC on LaSOT/TNL2K under 0-tensor setting, and 63.9%/49.8% to 53.4%/41.1 under template (i.e., visual pooling feature) setting. Likewise, without language for tracking, the performance of VLT_TT drops from 66.3%/52.2% to 60.7%/48.2% in SUC on LaSOT/TNL2K under 0-tensor setting, and 67.3%/53.1% to 61.0%/49.1% under template setting. All this reveals the importance of linguistic cues for tracking and shows that the learned representations are indeed multi-modal representations.
>
> We thank the reviewer again and will include all the results with analysis in revision.
>
> **Table #3**: Ablation experiments of using a 0-tensor or a visual pooling feature (i.e., template in the table) for tracking.
> | # | Method | Setting | Language | LaSOT | LaSOT | TNL2K | TNL2K |
> | :-: | :-: | :-: | :-: | :-: | :-: | :-: | :-: |
> |  |  |  |  | SUC (%) | P (%) | SUC (%) | P (%) |
> | 1 | VLT_SCAR | 0-tensor | w/o. language (i.e., inference with 0-tensor only) | 50.8 | 52.6 | 39.5 | 41.2 |
> | 2 | VLT_SCAR | 0-tensor | w/. language | 65.2 | 69.1 | 48.3 | 46.6 |
> | 3 | VLT_SCAR | template | w/o. language (i.e., inference with template only) | 53.4 | 54.6 | 41.1 | 42.9 |
> | 4 | VLT_SCAR | template | w/. language | 63.9 | 67.9 | 49.8 | 51.1 |
> | |
> | 5 | VLT_TT | 0-tensor | w/o. language (i.e., inference with 0-tensor only) | 60.7 | 63.1 | 48.2 | 46.8 |
> | 6 | VLT_TT | 0-tensor | w/. language | 66.3 | 70.5 | 52.2 | 52.1 |
> | 7 | VLT_TT | template | w/o. language (i.e., inference with template only) | 61.0 | 63.4 | 49.1 | 48.3 |
> | 8 | VLT_TT | template | w/. language | 67.3 | 72.1 | 53.1 | 53.3 |
>
> ***

---

> ### Author Response · Authors · 2022-08-01
> **Response to Reviewer YvUr (part 2)**
>
> ***Q2: (1) Modality Mixer is too simple to be the main contribution. It’s cross-modal channel attention. The authors attempted to make the structure more complex by designing a residual connection. (2) Ablation analysis is required for the Modality Mixer.***
>
> **A2**: **(1)** We have somehow different opinions on the statement of "Modality Mixer is too simple to be the main contribution". First, it is not "too simple". Indeed, the Modality Mixer (ModaMixer) contains three components, including (i) cross-modal channel attention, (ii) asymmetrical searching strategy (ASS) and (iii) residual connection. The cross-modal channel attention is used to interact two modalities, the ASS to search adaptive structures for different modalities, and the residual connection to enhance fused representation. Although the implementations of these components seem simple, the idea or insight inside the ModaMixer is NOT simple, which is evidenced by its effectiveness in improving tracking performance.
>
> In addition, the usage of "residual connection" is not intended to make the structure of ModaMixer more complex. Instead, it is designed to avoid losing informative vision details and mitigate the noise or bias that may exist in language description, with the goal of further improving performance. In fact, many well-known works such as Transform [*1] utilize the similar idea for the same purpose.
>
> We thank the reviewer for this comment and will add clarification in revision to make it more clear.
>
> [*1] Vaswani et al. Attention is all you need. In NeurIPS 2017.
>
> **(2)** We appreciate this insightful comment. As suggested, we have conducted ablation experiments for the ModaMixer. In specific, we analyze the cross-modal channel attention and the residual connection in ModaMixer. The experiments are conducted with the "template" setting and results are shown in Tab. #2.
>
> From Tab. #2, we can see that, when using only cross-modal channel attention (i.e., VLT_SCAR w/o Residual Connection), the performance is increased by 9.0%/7.0% from 52.1%/40.7% to 61.1%/47.7% in SUC on LaSOT and TNL2K, showing the effectiveness of multimodal fusion. In addition, when adding residual connection (i.e., VLT_SCAT by default), the performance is further improved by 2.8%/2.1% from 61.1%/47.7% to 63.9%/49.8%, which verifies the importance of residual connection in ModaMixer. Based on this ablation analysis, we argue that final improvement by ModaMixes can be attributed to both multimodal fusion and the usage of residual connection, along with ASS (see ablation experiment in Tab. 3 of the manuscript). We will include the ablation experiments with analysis in revision. Again, thanks.
>
>
> **Table #2**: Ablation studies on ModaMixer.
> | # | Method | Setting | LaSOT | LaSOT | TNL2K | TNL2K |
> | :-: | :-: | :-: | :-: | :-: | :-: | :-: |
> |  |  |  | SUC (%) | P (%) | SUC (%) | P (%) |
> | 1 | VLT_SCAR | w/o. Cross-modal Channel Attention and Residual Connection  | 52.1 | 50.6 | 40.7 | 40.2 |
> | 2 | VLT_SCAR | w/o. Residual Connection | 61.1 | 63.6 | 47.7 | 48.1 |
> | 3 | VLT_SCAR | default | 63.9 | 67.9 | 49.8 | 51.1 |
>
> ***

---

> ### Author Response · Authors · 2022-08-01
> **Response to Reviewer YvUr (part 1)**
>
> We thank the reviewer for providing valuable and thoughtful comments on our work. We provide our responses below to address the reviewer’s concerns, and remain committed to clarifying further questions that may arise during the discussion period.
>
> ***
> ***Q1: Different training data for the proposed trackers and baselines.***
>
> **A1**: Thanks for this helpful comment. As suggested, we retrain VLT_SCAR and VLT_TT and their baselines with the same data setting to ensure a fair comparison. The results are listed in Tab. #1:
>
> (1) We retrain SiamCAR with the same data setting of VLT_SCAR (#3 vs #5). Compared to the default setting (#1), double data volume and three more data sources contain different biases, which affect the trained model to produce biased outcomes, as illustrated in [*1-*3]. From the results (#3), the addition of TNL2K training set significantly improves the default #1 with 4.4% gains in SUC on TNL2K, whereas the performance on LaSOT slightly decreases. Compared to VLT_SCAR with the same setting (#5), #3 is still suppressed for 8.6%/7.4% of SUC and P on TNL2K, respectively.
>
> (2) We also retrain VLT_SCAR with the only LaSOT training set (#4), which keeps aligned with #2. The SUC scores on LaSOT and TNL2K degrade heavily to 57.0%/39.0% compared to the default #5, respectively. **This is caused by the great reduction of the language-annotated training data, i.e., from 1120(LaSOT)+1300(TNL2K)+9335(GOT-10k) to 1120(LaSOT).** Our model is hard to learn a good multimodal representation with the quite less language-annotated training data, which violates our intention. Even though, our VLT_SCAR (#4) still outperforms the baseline SiamCAR (#2) for 5.4%/4.0% of SUC scores on LaSOT/TNL2K.
>
> (3) TransT is also retrained with the same data setting as VLT_TT (#7 vs #8). More data sources bring similar biases and influence the performance as SiamCAR, compared to default TransT (#6). Our VLT_TT (#8) still achieves superior scores on both LaSOT and TNL2K.
>
> We will include the results and comparison in revision. Thanks.
>
> **Table #1**: Comparison with different data volumes and sources.
> | # | Method | Data Volume | Data Source | LaSOT | LaSOT | TNL2K | TNL2K |
> | :-: | :-: | :-: | :-: | :-: | :-: | :-: | :-: |
> |  |  |  |  | SUC (%) | P (%) | SUC (%) | P (%) |
> | 1 | SiamCAR | 60W×20Epoch | VID, YTBB, DET, COCO | 50.7 | 51.0 | 35.3 | 38.4 |
> | 2 | SiamCAR | 60W×20Epoch | LaSOT | 51.6 | 52.3 | 35.0 | 36.4 |
> | 3 | SiamCAR | 120W×20Epoch | VID, YTBB, DET, COCO, GOT-10K, LaSOT, TNL2K | 48.7 | 46.6 | 39.7 | 39.2 |
> | 4 | VLT_SCAR | 60W×20Epoch | LaSOT | 57.0 | 58.6 | 39.0 | 39.8 |
> | 5 | VLT_SCAR | 120W×20Epoch | VID, YTBB, DET, COCO, GOT-10K, LaSOT, TNL2K | 63.9 | 67.9 | 48.3 | 46.6 |
> | |
> | 6 | TransT | 3.8W×1000Epoch | COCO, GOT-10K, LaSOT, TrackingNet | 64.9 | 69.0 | 50.7 | 51.7 |
> | 7 | TransT | 3.8W×1000Epoch | COCO, GOT-10K, LaSOT, TrackingNet, TNL2K | 62.2 | 65.2 | 51.2 | 52.3 |
> | 8 | VLT_TT | 3.8W×1000Epoch | COCO, GOT-10K, LaSOT, TrackingNet, TNL2K | 67.3 | 72.1 | 53.1 | 53.3 |
>
> [*1] Chang et al. Active bias: Training more accurate neural networks by emphasizing high variance samples. In NeurIPS 2017.
>
> [*2] Kim et al. Learning not to learn: Training deep neural networks with biased data. In CVPR 2019.
>
> [*3] Mehrabi et al. A survey on bias and fairness in machine learning. In CSUR 2021.
>
> ***

---

> ### Author Response · Authors · 2022-08-06
> **Discussion invitation**
>
> We thank you for the precious review time and valuable comments. We have provided corresponding responses and results, which we believe have covered your concerns. We hope to further discuss with you whether or not your concerns have been addressed. Please let us know if you still have any unclear parts of our work.

---

### Official Review · Reviewer_5MXU · 2022-07-11

**Rating:** 8
**Confidence:** 5
**Soundness:** 4 excellent
**Presentation:** 4 excellent
**Contribution:** 4 excellent

**Summary:**

This paper proposes an interesting solution to show how to achieve state-of-the-art (sota) tracking performance without relying on the complex Transformer architecture in visual tracking. Specifically, the authors explore a unified multimodal learning of vision and language under the simple ConvNets for tracking. The authors first give an in-depth analysis of the limitations of previous solutions and then introduce an innovative unified framework that learns better vision-language (VL) representation by proposing ModaMixer and asymmetrical searching strategy (ASS). The former well enhances the interaction inside vision-language for discriminative representation, while the later improves the adaption of the learned VL representation by ModaMixer. Extensive experiments on five challenging datasets demonstrates that, the proposed method improves a pure CNN-based baseline to be competitive and even better than recent Transformer-based counterparts. Moreover, the proposed method is general and can also be applied to improve Transformer tracking architecture, as shown by experiments. In addition, some theoretical analysis is given to explain the effectiveness of the proposed method.

**Questions:**

See Weaknesses.

**Limitations:**

I don’t see the discussion related to the limitations and broader impact. I hope the author can provide a brief discuss on this part in the response and are encouraged to detail this part in the revision.

**Strengths And Weaknesses:**

Strengths

Overall, I am rather positive on this paper. In particular, I really like the motivation of this work that aims at finding alternative path, instead of relying on Transformer architecture (although it is shown to be powerful), to achieve sota tracking by exploring the almost ignored multimodal learning, which I believe can inspire many other works on tracking and facilitates this field. The strengths in this work include:

(1) Novelty. This paper introduces an innovative framework of multimodal vision-language learning (VL) for object tracking. The proposed ModaMixer enables a good interaction between the two modalities throughout the whole network, which is completely different from other works that only interact two modalities at the result fusion stage. In addition, another novel point is the proposed ASS that adapts the representation of different branches and modalities in an asymmetrical way. Compared with the conventional symmetrical way, ASS is the first time to show the asymmetrical architecture may be more suitable for representation learning in tracking and may provide new insights to future research. I like these novel ideas in this work.

(2) Good performance. This paper demonstrates excellent performance in improving the simple pure CNN-based tracking baselines. In specific on the challenging LaSOT, the proposed approach improves the baseline from 50.7% to 65.2% in SUC with absolute 14.5% gains, which is competitive and even better than recent Transformer-based trackers. Compared to previous best VL method with 54.0% SUC, this paper shows 11.2% improvement. The performance on other benchmarks is also sota. This excellent performance clearly shows the effectiveness and advantages of the proposed method.

(3) Generality. The proposed method is general and not limited to CNN-based framework. The authors show this point by applying their methods on a Transformer-based tracking framework and show promising improvements on all the five challenging benchmarks, which is consistent with the improvements shown for CNN-based framework.

(4) Rich experiments and analysis. The authors provide extensive experiments and analysis for the proposed method. I appreciate this. The experimental analysis with various ablation studies allows a better understanding of each module and overall performance, and the theoretical analysis gives an explanation why the proposed method works well in improving tracking performance.

(5) Good writing and organization. This paper is well written and organized. Each section has a clear motivation. It’s easy to follow the ideas. I enjoy reading the paper.

Overall, I believe this paper is significant to the visual tracking community because it shows new insights and directions in designing simple but effective tracking framework with sota performance.

Weaknesses

Although this paper is technically sound and novel, I have some minor concerns or questions.
(1) In this work, language is crucial. I noticed that in the experiment the pseudo language description generated by an image caption model does not show significant improvements. What do you think are the reasons causing this?
(2) The proposed method is shown to be general, which is nice. However, it will be great if the authors can show more examples such as conducting more experiments on additional pure CNN-based frameworks like SiamRPN++ (Li et al, CVPR, 2019).
(3) The authors show many comparisons with other sota methods. But I don’t see qualitative comparison with other approaches. It will be nice to see these results and comparison to other sota methods.
(4) For experiments with partial language (50% and 75%) in the supplementary material, how do you determine the which 50% or 75% should be used? Randomly generating? If you randomly generate it, it will be better to do multiple (e.g., 3) times of experiments.

---

> ### Author Response · Authors · 2022-08-01
> **Response to Reviewer 5MXU**
>
> We appreciate your careful review and thoughtful comments. We are encouraged and grateful that the reviewer found our approach to be well-motivated, neat and inspiring. Below, we address the concerns that were raised.
>
> ***
> ***Q1: Why does not the pseudo language description generated by an image caption model show significant improvements?***
>
> **A1**: Thanks for this insightful comment. The reason lies in the domain gap between tracking datasets and existing image caption datasets, which results in poor quality of the generated language description by image caption model (e.g., [*1]) for tracking. For example, the official annotation for the first row in Fig. 7 of the supplementary material is "black bird standing on the ground", while the caption model [*1] generates the language description "A blurry image of a person and a cat", which is inaccurate in describing the scene. As a consequence, the usage of such inaccurate language description by the image caption model may introduce noise into training and testing of the tracker, leading to inferior performance. To make it clear, we will add clarification on this point in revision. Thanks, again.
>
> ***
> ***Q2: More examples such as conducting more experiments on additional pure CNN-based frameworks like SiamRPN++?***
>
> **A2**: Thanks for this constructive suggestion. As suggested, we apply our method on SiamRPN++ to develop the new VL tracker VLT_RPN++ and show the comparison in Tab. #1. As shown in Tab. #1, compared with the baseline SiamRPN++, the proposed VLT_RPN++ achieves considerable SUC gains of 9.4%/4.5% on LaSOT/TNL2K, respectively, which demonstrates the effectiveness and generalization of our multimodal VL representation in improving tracking. We will include the results and comparison in revision. Thanks.
>
> **Table #1**: Comparison bettwen VLT_RPN++ and its baseline SiamRPN++.
> | # | Method | LaSOT | LaSOT | TNL2K | TNL2K |
> | :-: | :-: | :-: | :-: | :-: | :-: |
> |  |  | SUC (%) | P (%) | SUC (%) | P (%) |
> | 1 | VLT_RPN++ | 59.0 | 62.6 | 45.8 | 47.4 |
> | 2 | SiamRPN++ | 49.6 | 49.1 | 41.3 | 41.2 |
>
> ***
> ***Q3: Add qualitative comparison with other sota methods.***
>
> **A3**: Thanks for this comment. As suggested, we update the supplementary material by adding qualitative results and comparison with other SOTA trackers in Fig. 6. Please kindly check the updated supplementary material for reference. Thank you.
>
> ***
> ***Q4: (1) For experiments with partial language (50% and 75%) in the supplementary material, how do you determine the which 50% or 75% should be used? (2) If you randomly generate it, it will be better to do multiple (e.g., 3) times of experiments.***
>
> **A4**: Sorry for the confusion. **(1)** For experiments with partial language, we generate the training date by sampling from each language-annotated datasets randomly based on the ratio setting. For example, for 50% language-annotated data, we randomly sample 50% of the data from each dataset. The procedure is the same for other settings. **(2)** Consider the randomness, the experiments are repeated for multiple times. We will clarify this in revision. Thanks.
>
> ***
>
> ***Q5: I don’t see the discussion related to the limitations and broader impact.***
>
> **A5**: Thanks for this helpful comment. One limitation of our work (also for other existing VL trackers) lies in the lack of language-annotated tracking datasets (only 1.7%), which is crucial to train an effective vision-language tracker. To remedy this issue, in our paper, we try to generate language descriptions for training videos with an image caption model [*1]. However, due to the large domain gap, the caption model shows unsatisfactory generation quality, which stops the step to enjoy benefits from huge scale multimodality training as in other VL tasks. In the follow-up work, we will address this problem.
>
>
> [*1] Radford et al. Learning transferable visual models from natural language supervision. In ICML 2021.
> ***

---

> > ### Comment · Reviewer_5MXU · 2022-08-10
> > **Post rebuttal comments**
> >
> > I highly value this work due to its novelty and promising improvements. I read other reviewers' comments and the rebuttal, and find that the authors have carefully and adequately addressed all my concerns. This work is the best tracking paper among ones that I have reviewed in NeurIPS. Thus, I keep my original rating as strong accept.

---

### Official Review · Reviewer_nvbZ · 2022-07-11

**Rating:** 6
**Confidence:** 4
**Soundness:** 3 good
**Presentation:** 3 good
**Contribution:** 3 good

**Summary:**

This paper proposes a tracking algorithm by designing a ModaMixer and asymmetrical networks to learn unified-adaptive VL representation. The combination of the two modules gives very good results. Applying language features to tracking tasks is a very reasonable and innovative multimodal learning algorithm.

**Questions:**

This paper doesn't clearly state how to get the language description of a template during inference. If the language description is from annotation or manual setting, it is not reasonable enough to compare results with algorithms that do not use language features.

**Limitations:**

The article does not include a discussion of solving common tracking problems such as target deformation and occlusion. The introduction of the template update module may have better results on some samples.

**Strengths And Weaknesses:**

Strengths
•	The paper is clearly written.
•	Results greatly improve the performance and seems achieve SOTA.
•	The paper has adequate ablation experiments.

Weaknesses
•	I noticed that the results in Table 2 used different settings(0-Tensor vs Template). The author just lists the best results, the results of same algorithms should come from the exact same network structure, otherwise, they should be shown separately. For the same reason, the result of TNL2K in table 3(48.3/46.6) is different from it in Table 2(49.8/51.0).

---

> ### Author Response · Authors · 2022-08-01
> **Response to Reviewer nvbZ (part 2)**
>
> ***Q3: If the language description (in Q2) is from annotation or manual setting, it is not reasonable enough to compare results with algorithms that do not use language features.***
>
> **A3**: Thanks for this insightful comment. The goal of our work is to demonstrate the power of language description (given only in the first frame of a video) in improving video object tracking. For this purpose, we compare the proposed VLT with (1) its CNN (or Transformer)-based baseline, (2) other CNN (or Transformer)-based state-of-the-art (SOTA) trackers, and (3) other vision-language (VL) trackers, as in Tab. 2 of the manuscript.
>
> The comparison with (1) is to show the performance improvement gained over the baseline method by leveraging linguistic information, which verifies the effectiveness of our method. The comparison with (2) is to demonstrate that our method can mitigate the huge gap between VL tracker and recent vision-only SOTAs, which shows its potential in further improving tracking. The comparison with (3) is to validate the effectiveness and superiority of our vision-language representation. Considering these reasons, we believe that it is necessary and reasonable to compare our method with various trackers, including its baseline, other vision-only and vision-language trackers, as in other VL trackers (e.g., SNLT [18]).
>
> We thank the reviewer again and will include the above explanation in revision.
>
> ***
> ***Q4: The article does not include a discussion of solving common tracking problems such as target deformation and occlusion.***
>
> **A4**: Thanks for the thoughtful comments. Indeed, in Section A.12 ("Attribute-based Performance Analysis") of the supplementary material, we show the improvements of our method in various challenges (or the so-called attributes), including deformation, occlusion, background clutter, viewpoint change, etc, owing to the robust representation learned by our approach. The detailed SUC scores are listed in Tab. #2. With the proposed multimodal representation learning, VLT_SCAR significantly improves the baseline SiamCAR for 15.9%/15.8%/16% on the attributes of deformation, partial occlusion and full occlusion. VLT_TT also brings 1.7%/3.0%/2.8% gains compared to the baseline TransT. We will include the results and more quantitative and qualitative analysis in revision to discuss how our method solves common tracking challenges. Thanks, again.
>
>
> **Table #2:** SUC scores (%) on LaSOT under attributes of Aspect Ration Change (ARC), Low Resolution (LR), Our-of-View (OV), Fast Motion (FM), Full Occlusion (FO), Scale Variation (SV), Viewpoint Change (VC), Background Clutter (BC), Rotation (Rot), Camera Motion (CM), Motion Blur (MB), Deformation (Def), Partial Occlusion (PO) and Illumination Variation (IV).
> | # | Method | ARC | LR | OV | FM | FO | SV | VC | BC | Rot | CM | MB | Def | PO | IV |
> | :-: | :-: | :-: | :-: | :-: | :-: | :-: | :-: | :-: | :-: | :-: | :-: | :-: | :-: | :-: | :-: |
> | 1 | VLT_SCAR | 61.8 | 54.6 | 53.0 | 45.8 | 53.4 | 63.7 | 60.6 | 58.6 | 64.0 | 65.2 | 60.7 | 66.3 | 61.5 | 65.1 |
> | 2 | SiamCAR | 45.9 | 38.5 | 41.0 | 31.7 | 37.4 | 48.4 | 43.9 | 42.2 | 46.9 | 51.7 | 46.4 | 50.4 | 45.7 | 53.3 |
> | 3 | SNLT | 51.8 | 46.7 | 46.4 | 38.4 | 42.6 | 53.8 | 54.2 | 49.2 | 52.6 | 57.6 | 51.8 | 55.5 | 49.3 | 60.6 |
> | |
> | 4 | VLT_TT | 65.6 | 59.0 | 58.9 | 50.6 | 58.1 | 67.1 | 65.6 | 62.0 | 67.2 | 68.4 | 65.7 | 68.7 | 65.0 | 64.1 |
> | 5 | TransT | 63.2 | 56.4 | 58.2 | 51.0 | 55.3 | 64.6 | 61.7 | 57.9 | 64.3 | 67.2 | 63.0 | 67.0 | 62.0 | 65.2 |
> | 6 | TrDiMP | 62.6 | 58.8 | 61.0 | 54.0 | 56.7 | 63.6 | 62.6 | 59.9 | 62.5 | 68.5 | 62.9 | 64.7 | 61.5 | 67.5 |
>
> ***
> ***Q5: The introduction of the template update module may have better results on some samples.***
>
> **A5**: We agree with the reviewer and appreciate this constructive suggestion. Incorporation of temporal cues can further improve the adaption of the tracker in target localization. However, to ensure fair comparison with our baseline trackers SiamCAR and TransT and other VL tracker SNLT (these three trackers do not use the template update), this update mechanism is currently not applied in our method. We leave this as the future work. Thanks.
>
> ***

---

> ### Author Response · Authors · 2022-08-01
> **Response to Reviewer nvbZ (part 1)**
>
> We appreciate the reviewer for the encouraging comments and constructive suggestions, and provide our responses below to address the reviewer's questions and concerns.
>
> ***
> ***Q1: The results of same algorithms should come from the exact same network structure, otherwise, they should be shown separately. For the same reason, the result of TNL2K in table 3(48.3/46.6) is different from it in Table 2(49.8/51.0).***
>
> **A1**: Thanks for this helpful comment. We show the detailed results of VLT_SCAR and VLT_TT under different settings (i.e., using "0-tensor" and "template", respectively), in Tab. #1. To make the results more comprehensive and consistent, as suggested, we will include the results in Tab. #1 into Tab. 2 of the manuscript in revision. Again, thanks.
>
> **Table #1:** Detailed results of VLT_SCAR and VLT_TT using "0-tensor" and "template", respectively.
> | # | Method | Setting | LaSOT | LaSOT | LaSOTExt | LaSOTExt | TNL2K | TNL2K | GOT | GOT | OTB99L | OTB99L |
> | :-:| :-: | :-: | :-: | :-: | :-: | :-: | :-: | :-: | :-: | :-: | :-: | :-: |
> |  |  |  | SUC (%) | P (%) | SUC (%) | P (%) | SUC (%) | P (%) | AO (%) | SR0.5 (%) | SUC (%) | P (%) |
> | 1 | VLT_SCAR | 0-tensor | 65.2 | 69.1 | 41.2 | 47.5 | 48.3 | 46.6 | 61.4 | 72.4 | 72.7 | 88.8 |
> | 2 | VLT-SCAR | template | 63.9 | 67.9 | 44.7 | 51.6 | 49.8 | 51.1 | 61.0 | 70.8 | 73.9 | 89.8 |
> | |
> | 3 | VLT_TT | 0-tensor | 66.3 | 70.5 | 45.4 | 52.1 | 52.2 | 52.1 | 68.4 | 81.5 | 74.7 | 91.2 |
> | 4 | VLT_TT | template | 67.3 | 72.1 | 48.4 | 55.9 | 53.1 | 53.3 | 69.4 | 81.1 | 76.4 | 93.1 |
>
>
> ***
> ***Q2: This paper doesn't clearly state how to get the language description of a template during inference.***
>
> **A2**: Sorry about this. The language description of the template comes from the annotation of the benchmark, which is the same as in other vision-language trackers (e.g., [17, 18] in the manuscript). Please note, the language description is only given in the first frame of a video. For datasets without official language description, we have designed two strategies as discussed in Tab. 5 (c) of the manuscript (e.g., not using language description or generating the language description with a caption model [*1]). We will clarify this point in the revision by adding more explanation. Thanks.
>
> [*1] Radford et al. Learning transferable visual models from natural language supervision. In ICML 2021.
>
> ***

---

### Meta-Review · Area_Chair_YgRS · 2022-08-30

**Recommendation:** Accept
**Confidence:** Certain

**Metareview:**

All three reviewers lean towards the acceptance of the paper. Reviewer YvUr was not 100% excited about the paper, pointing out the simplicity of the approach and lacking ablations. We encourage the authors to include the new materials they prepared for the rebuttal in the final version of the paper.

**Award:**

No

---

### Decision · Program_Chairs · 2022-09-14

Accept